# Formulation, Characterization, and In Vitro/In Vivo Efficacy Studies of a Novel Liposomal Drug Delivery System of Amphiphilic Jaspine B for Treatment of Synovial Sarcoma

**DOI:** 10.3390/md20080509

**Published:** 2022-08-10

**Authors:** Sana Khajeh pour, Sameena Mateen, Srinath Pashikanti, Jared J. Barrott, Ali Aghazadeh-Habashi

**Affiliations:** College of Pharmacy, Idaho State University, Pocatello, ID 83209, USA

**Keywords:** liposomes, jaspine B, microfluidic method, drug delivery, synovial sarcoma

## Abstract

Sphingomyelin is a cell membrane sphingolipid that is upregulated in synovial sarcoma (SS). Jaspine B has been shown to inhibit sphingomyelin synthase, which synthesizes sphingomyelin from ceramide, a critical signal transducer; however, jaspine B’s low bioavailability limits its application as a promising treatment option. To address this shortcoming, we used microfluidics to develop a liposomal delivery system with increased anticancer efficacy. The nano-liposome size was determined by transmission electron microscopy. The jaspine B liposome was tested for its tumor inhibitory efficacy compared to plain jaspine B in in vitro and in vivo studies. The human SS cell line was tested for cell viability using varying jaspine B concentrations. In a mouse model of SS, tumor growth suppression was evaluated during four weeks of treatment (3 times/week). The results show that jaspine B was successfully formulated in the liposomes with a size ranging from 127.5 ± 61.2 nm. The MTT assay and animal study results indicate that jaspine B liposomes dose-dependently lowers cell viability in the SS cell line and effectively suppresses tumor cell growth in the SS animal model. The novel liposome drug delivery system addresses jaspine B’s low bioavailability issues and improves its therapeutic efficacy.

## 1. Introduction

As, a marine sponge extract and anhydrophytosphingosine, jaspine B, inhibits sphingosine kinases and their proliferative activities [1]. Since its isolation, several biochemical studies have demonstrated its ability to inhibit microtubule formation, cell growth, and cell death program stimulation [2]. Human Sk-Mel28 melanoma cell lines and Murine B16 proliferation, for example, are inhibited by jaspine B in a time- and dose-dependent manner. Exposing these cells to jaspine B triggers cell death by typical apoptosis, as indicated by phosphatidylserine externalization, the release of cytochrome c, and caspase processing [2]. Although biological studies of jaspine B and its congeners have exhibited cytotoxic properties in several cancer cell lines [3,4,5], no commercial jaspine B natural products are available for therapeutic use.

Sphingomyelin, a cell membrane sphingolipid, plays a vital role in cell growth and mitosis and is upregulated in osteosarcoma and synovial sarcoma (SS). The metabolic intermediate of the sphingolipid pathway, ceramide, is critical in cellular signal transduction and apoptotic pathways. Jaspine B inhibits sphingomyelin synthase [2] (Figure 1) and induces apoptosis in HeLa cells, which are associated with disrupting sphingolipid homeostasis, resulting in increased ceramide levels [3]. In gastric epithelial cells, jaspine B targets the autophagy pathways and demonstrates anticancer effects [4]. The bioavailability and accumulation of jaspine B in the cells during the steady state is a significant factor that correlates with its cytotoxic efficacy [6]. However, low bioavailability due to limited intestinal absorption and extensive oral clearance impedes its application as a promising anticancer agent. Coadministration with bile salts increases oral bioavailability and may enhance its pharmacological effects [7].

This study developed a liposomal delivery system to improve jaspine B’s pharmacokinetics. We also tested its ability to inhibit cell proliferation in the sarcoma cell line and SS tumor growth in the animal model. Liposomes, a type of vesicle used for biomedical and pharmaceutical applications, are biocompatible delivery systems that enhance a drug’s oral bioavailability [8]. Liposomes improve hydrophobic compound solubility and prevent chemical or enzymatic degradation in the gut. The lipid bilayer structure provides other advantages, such as cell membrane adherence, enhanced permeability, and lymphatic uptake. Liposomes consist of a phospholipid bilayer enclosing a small volume of aqueous buffer. Their size may vary from tens of nanometers to hundreds of micrometers, based on the protocol utilized for their preparation and use. Liposomes have successfully improved the oral bioavailability of various compounds, including peptides, proteins, and hydrophilic and lipophilic drugs [9]. Liposomes are formulated using several methods. A new microfluidic fabrication uses streams of an ethanolic solution and an aqueous buffer and lipids forced through the central channel of a microfluidic mixer cartridge. (Figure 2). The two liquid phases pass through a thin sheet with rectangular cross sections to form vesicular liposomes. Vesicle size is controlled by adjusting the total flow rate (TFR) and flow rate ratio (FRR) between the lipid and aqueous phases [10].

We developed a novel jaspine B liposomal delivery system of jaspine B that enhance both in vitro and in vivo efficacy. We characterized the liposomal formulation using transmission electron microscopy (TEM) and LC-MS/MS methods and tested the in vitro and in vivo efficacy in an SS cell line and an animal model.

## 2. Results

### 2.1. Stereoselective Scale-Up Synthesis of Amphiphilic Jaspine B

Jaspine B was synthesized, and its scale-up was performed successfully with more than 80% yield from its precursor, according to our published method [11].

Its chemical structure and purity were confirmed by characteristic peaks in the 1H- NMR spectrum (400 MHz, CD3 OD): δ = 0.88 (t, 3 H), 1.20–1.51 (m, 24 H), 1.61–1.73 (m, 2 H), 3.67–3.74 (m, 1 H), 3.76–3.84 (m, 1 H), 3.84–3.95 (m, 2 H), 4.25 (br s, 1 H). (Figure 3) and LC-MS/MS chromatogram [7] (Figure 4A). The scale-up jaspine B characterization is in agreement with the previously reported study [11].

### 2.2. LC-MS/MS Analysis of Jaspine B in Liposomes

A published LC-MS/MS method [7] was successfully validated and adopted for jaspine B quantification. Figure 4 depicts the chromatogram generated by Electron Spray Ionization (ESI) in positive mode and MRM at *m*/*z* 299.8→270.1 for jaspine B and *m*/*z* 336.1→320.0 for berberine as internal standard (IS). The retention times of jaspine B and IS were 5.24 and 5.70 min in solution and 4.78 and 5.17 in the tested samples. The slight difference in the retention times resulted from a marginally altered solvent composition used for reconstitution. However, the mass and fragment transitions were identical for these compounds in the standard solution (Figure 4A) or liposomal extract (Figure 4B).

### 2.3. Jaspine B Liposome Characterization

Different jaspine B concentrations and instrument parameters (FRR) were tested for liposome encapsulation efficiency (EE) and size determination. Phospholipid bilayer visualization occurs due to phosphorus atom contrast under electron microscopy [10]. The impact of different FRRs and jaspine B concentrations at an 8 mL/min TFR on the size distribution and EE% are presented in Table 1. Figure 5 depicts different batches of formulated liposomes at variable FRR and jaspine B concentrations with some distinct outer and inner layers visualization (Figure 5C).

Table 1 and Figure 5C results indicate that EE% was highest (97%) when FRR was set at 2:1 and jaspine B concentration at 2 mg/mL, while the size of distribution of liposomes was at 127.5 ± 61.2 (Figure 5E). Considering an EE of 97%, the drug: lipid mass ratio was 0.39.

Jaspine B liposomes were stable for more than two months in PBS pH 7.4 at 4 °C; their size and EE% did not significantly change.

### 2.4. In Vitro Human SS Cells Viability Assay

Using the MTT assay and Yamato-SS cell line, the jaspine B IC_50_ and its liposome were determined after 72 h of exposure to a drug-free medium, empty liposome, or the final concentrations, ranging from 0.01 to 100 µM of jaspine B or equivalent concentrations of jaspine B liposome (Figure 6).

The IC_50_ and log IC_50_ values for both jaspine B and its liposome in the Yamato-SS cell line are presented in Table 2. The IC_50_ value (mean ± SD) for jaspine B was 0.36 ± 0.07 µM, which improved when jaspine B was encapsulated in liposomes (0.06 ± 0.01 µM).

### 2.5. In Vivo Efficacy Study in SS Mouse Model

Jaspine B’s liposomal formulation in vivo anticancer effect was investigated using an SS animal model. Figure 7 shows the tumor size fold change compared to the control group during the four weeks of treatment (3 doses/week). Mice were monitored regularly, and no outward signs of toxicity were evident due to different treatments. Jaspine B and its liposomal formulation suppressed tumor growth. This effect was more pronounced in the jaspine B liposome-treated group. A set of representative images, one mouse per group, is depicted in Figure 8.

## 3. Discussion

Yamato- SS cell line study results indicate that jaspine B formulation in a liposome significantly increased in vitro anticancer potency, with a lower IC_50_ than jaspine B (Figure 6 and Table 2). Jaspine B’s efficacy against tumors originating from various tissues has been reported, where its effectiveness was dependent upon steady-state cellular accumulation [12]. Our observation may also be attributed to improved jaspine B stability and liposomal lipid bilayer impact on enhanced cell permeation and accumulation.

Systemic administration of jaspine B significantly reduced lung metastatic melanoma cell growth [13]. However, extensive systemic clearance or low bioavailability after oral administration, due to poor solubility and extensive first-pass effect, hampered its efficacy and reduced suitability for cancer treatment. Choi et al. used a bile salts coadministration approach. They improved jaspine B bioavailability by several folds, attributed to jaspine B’s permeation through the lipophilic cell membrane and tight junction [7]. These results suggest that developing oral jaspine B formulations with bile acids and phospholipids could increase the compound’s bioavailability and tissue permeability [14]; however, the observed pharmacokinetic improvement was not evaluated for pharmacodynamic effect. In other studies, the administration of several anticancer drugs, such as doxorubicin [15] and paclitaxel [16], in phospholipid mixture formulations also resulted in a higher cellular concentration and reduced systemic toxicity compared to the plain drug.

Similarly, we observed a pronounced pharmacodynamic effect on tumor growth suppression in a SS animal model after 4-week oral liposomal delivery of jaspine B, which outperformed plain jaspine B (Figure 7 and Figure 8). A significant level of deference was not attained due to the small number of animals enrolled in the study of jaspine B (*n* = 4) and its liposomal formulation (*n* = 3); however, we assume the observed in vivo efficacy improvement is rooted in jaspine B’s enriched oral bioavailability, parallel to the Choi et al. study. We are currently conducting a pharmacokinetic study in healthy animals to confirm the observed data and further elaborate on the mechanisms responsible for the observed effects.

The lipid compositions, their ratios, and jaspine B concentrations were determined from pilot study results, jaspine B physicochemical characteristics, the manufacturer’s protocol, initial suggestions from Precision NanoSystems scientists, and previous lipophilic molecule studies [17,18,19]. The lipid concentration was kept constant at 5 mg/mL because higher concentrations clogged the cartridge and did not yield any liposomes. The formulation and characterization data of jaspine B (Figure 5 and Table 1) indicate that an optimal liposome size with narrower size distribution and highest EE% was achieved using a lipid mixture of 2:4:4 *w*/*w* (cholesterol, DSPC, and DSPE-PEG), jaspine B concentration of 2 mg/mL, FFR of 2:1 and TFR of 8 mL/min. Although using a lower concentration of jaspine B at both values of FRR produced smaller liposomes, the size distribution was much broader, and EE% was much lower. Similar results were observed when a higher concentration of jaspine B with an FRR of 3:1 was used. The morphologic TEM image of liposomes clearly shows a lipidic bilayer structure surrounding an aqueous phase at the center. Jaspine B’s amphiphilic chemical structure (Figure 9) enabled its incorporation into the liposome’s lipidic bilayer shell, which enhanced vesicle formation, increased membrane permeability, and improved in vitro and in vivo efficacy.

The short-period stability study of the liposomes showed an optimal shelf-life for this nanocarrier delivery system. Further investigation is needed to evaluate the possibility of preparation and storage by other means (e.g., lyophilized powder), which could further extend its shelf-life and ease its administration and usage. Considering jaspine B’s remarkable anti-tumor effect, the proposed liposomal drug delivery system offers a potentially effective treatment for SS and other cancers.

## 4. Materials and Methods

### 4.1. Materials

Cholesterol, 1,2-distearoyl-sn-glycero-3-phosphocholine (DSPC), and DSPE-PEG2000 Carboxylic Acid were purchased from Avanti Polar Lipids (Alabaster, AL, USA). All other reagents used were analytical grade and purchased from Sigma Aldrich (St. Louis, MO, USA).

### 4.2. Stereoselective Scale-Up Synthesis of Amphiphilic Jaspine B towards Liposome Formation

Jaspine B was synthesized and scaled up based on our published method [11]. Briefly, stereoselective synthesis of jaspine B was performed using a chiral pool strategy starting with L-Serine (Figure 9). The second-generation chiral aminobutenolide is synthesized in gram quantities starting from L-Serine (Figure 9a). Chiral aminobutenolide (Figure 9b) has inherent chirality with unsaturated lactone (Michael acceptor) and protected nucleophiles (OH, NH_2_). Acetonide deprotection of aminobutenolide followed by Michael addition affords thermodynamically favorable enantiopure bicyclic furafuranone (Figure 9c) as the only product. The all-syn tri-substitution is achieved in this transformation. Functional group transformation of the lactone in bicyclic furafuranone (Figure 9c) provides an *N, O*-protected jaspine B precursor (Figure 9d). Global deprotection of jaspine B precursor under acidic conditions results in enantiopure jaspine B as HCl salt (Figure 9e).

Jaspine B’s structural core has an all-syn trisubstituted tetrahydrofuran core, which constitutes the C-2 lipophilic tetradecyl alkyl chain and C-3 hydroxy, C-4 amino polar functionalities. This substitution pattern differentiates jaspine B with a hydrophilic head group and a lipophilic tail, mimicking the sphingolipid-type scaffold. This characteristic differentiation makes jaspine B amphiphilic. It is noteworthy that the final step of the jaspine B synthesis involves global deprotection of *N-Cbz*, *N* and *O* pentanonide in an aqueous HCl under vigorous stirring and reflux conditions. Jaspine B’s amphiphilic nature appears as an oil droplet immiscible in an aqueous environment supporting the structural features of liposomal formation. Upon cooling, it solidifies as white solid immiscible HCl salt.

For structural characterization, using the Bruker DRX 400 Mhz spectrometer, NMR spectra were recorded with the chemical shifts (δ) reported in ppm relative to CD_3_OD (for ^1^H) as the internal standard. The AB Sciex (Foster City, CA, USA) QTRAP 5500 quadrupole mass spectrometer in positive electrospray ionization mode (ESI) was used for molecular ion and purity confirmation.

The chemical composition of this liposome involves Cholesterol, 1,2-distearoyl-sn-glycero-3-phosphocholine (DSPC), and DSPE-PEG2000 Carboxylic Acid. The amphiphilic jaspine B matches the other constituents of liposome and efficiently enhances the vesicle formation (Figure 10).

### 4.3. Preparation and Characterization of Liposomal Vesicles Using Microfluidics

Liposomes were prepared using a microfluidic mixing method using the NanoAssemblr™ Platform (Precision Nanosystems, Vancouver, BC, Canada). The mixer contains channels for organic and aqueous phases. Disposable syringes of 1 mL were used for inlet streams. Controlled TFR (8 mL/min) and aqueous to organic phase FRR (3:1 and 2:1) and two concentrations of 1 and 2 mg/mL jaspine B were used to achieve the optimum liposome size, encapsulation efficiency, and loading capacity. The temperature was controlled by a heating block unit and maintained at 60 °C.

To formulate the liposomes, 10 mg of lipid mixture (2:4:4 *w*/*w* of cholesterol, DSPC, and DSPE-PEG) was dissolved in 2 mL ethanol to achieve a 5 mg/mL lipid solution. Such lipid composition, ratio, and concentration have been successfully used in studies for liposomal encapsulation of lipophilic and hydrophilic molecules [17,18,19]. The lipid concentration was kept constant at 5 mg/mL to avoid cartridge clogging at higher concentrations.

Empty vesicles were manufactured by injecting lipids and phosphate-buffered saline (PBS 10 mM, pH 7.4) into the instrument’s separate chamber arms. The final liposomal formulation was collected from the chamber outlet and dialyzed at room temperature using PBS 10 mM. Then, liposome solutions were centrifuged in 10 kDa molecular weight cut-off falcon tubes for one hour at 4 °C to remove the remaining solvent. Jaspine B liposome formulation procedure was similar and used the same TFR and FRR input conditions, while jaspine B was dissolved in ethanol along with lipids due to its lipophilicity.

Size measurements and recording of the morphology of liposomes were performed using a TEM instrument according to a method described previously [20]. A total of 20 µL of the sample was deposited onto mesh carbon-coated copper grids to obtain a thin film. The excess solvent was drained gently by blotting filter paper during the verification process. Then, the grids were air-dried at room temperature. The prepared grids were transferred to a grid holder and examined by a TEM instrument (Gatan Inc., Pleasanton, CA, USA) at an accelerating voltage of 80 kV.

### 4.4. Jaspine B Analysis Using Liquid-Chromatography Mass Spectrometry (LC-MS/MS)

The level of jaspine B was measured using a validated LC-MS/MS method [7] in multiple reaction monitoring (MRM). The system was composed of liquid chromatography in tandem with mass spectrometry (Shimadzu, Columbia, MD, USA) with a controller (CBM-20A), two binary pumps (LC-30AD), an autosampler (SIL-30AC), and an AB Sciex (Foster City, CA, USA) QTRAP 5500 quadrupole mass spectrometer in positive electrospray ionization mode (ESI). The chromatograms were monitored and integrated by the Analyst 1.7.2 software (AB Sciex, Foster City, CA, USA).

LC separation was performed on an analytical reversed-phase column Kinetex^®^C18 100 × 2.1 mm (1.7 μm) (Phenomenex, Torrance, CA, USA) by a combination of 0.1% formic acid in acetonitrile and water (85:15, *v*/*v*) in isocratic mode at a flow rate of 0.2 mL/min.

The positive ion ESI mass spectrometric parameters were as follows: capillary voltage; 5.5 kV, temperature; 250 °C, declustering potential (DP); 50 V, and collision cell exit potential (CXP); 15 V quantification was performed using MRM at *m*/*z* 299.8→270.1 (jaspine B), and *m*/*z* 336.1→320.0 (IS). Nitrogen was used as collision gas, and the collision energies were set at 30–40 eV. Calibration curves using peak height ratio (analyte over IS) were constructed over the range of 0.5 to 16 ng/mL in liposomes to measure jaspine B concentrations in liposomes.

### 4.5. Encapsulation Efficiency of Jaspine B Liposomes

The EE% of jaspine B in the liposomes was determined using a previously described method with minor modification [21]. The liposome samples were centrifuged at 8000× *g* for 10 min to remove unencapsulated jaspine B; supernatants were treated with an equal volume of triton-X100 (Ameresco. Solon, OH, USA), followed by centrifugation at 20,000× *g* for 30 min. Twenty µL of 200 ng/mL IS was added to each sample, and the peak intensity was measured using an LC-MS/MS method. The content of jaspine B loaded in the liposomes was then calculated by a calibration curve. The EE was calculated using the following equation:EE (%) = MJL/MJI × 100(1)
where MJL is the mass of jaspine B loaded into the liposomes, and MJI is the initial mass of the jaspine B in the system.

The stability of jaspine B liposome solution in PBS pH 7.4 was tested for size and EE% after storage at 4 °C for two months.

### 4.6. In Vitro Cell Viability Assay

A cell viability assay was performed using a previously published method [22]. Dr. Torsten Nielson, University of British Columbia, provided the human Yamato-SS cell line (CVCL_6C44)which was maintained in Dulbecco’s modified Eagle medium (DMEM) with 10% fetal bovine serum (FBS) and 1% penicillin/streptomycin at 37 °C and 5% CO_2_. The in vitro cell viability assay was performed using MTT (3-(4, 5-dimethyl thiazolyl-2)-2,5-diphenyltetrazolium bromide) reagent to compare the viability and anti-proliferation effects of jaspine B with jaspine B liposome in different concentrations. Cell viability was normalized using a drug-free medium or empty liposome as a control treatment. Cells were seeded in a 96-well tissue culture dish at 10,000 cells/well and incubated for twenty-four hours. Serial dilutions of jaspine B and jaspine B liposomes were prepared so that the final concentration ranged from 0.01 to 100 µM. After 72 h of treatment, 5 mg/mL MTT was added to each well, resulting in a final concentration of 0.5 mg/mL MTT; cells were incubated at 37 °C and 5% CO_2_ for 2.5 h. Formazan crystals were resolubilized in 10% SDS and 0.01 M HCl, and the absorbance of the samples was measured at 570 nm and normalized to 650 nm using a microplate reading spectrophotometer, VarioscanLux (Thermofisher, Waltham, MA, USA). Each experiment was performed in triplicate.

### 4.7. SS Induction, Dosage Regimens, and In Vivo Study in Mice

An animal model of SS was used to further investigate and compare the efficacy of jaspine B liposomes with plain jaspine B. Mouse experiments were conducted with the approval of the Idaho State University’s Institutional Animal Care Committee under legal and ethical standards established by the National Research Council and published in the Guide for the Care and Use of Laboratory Animals (protocol #757, approved 5 July 2019). The previously described Rosa26-LSL-SS18-SSX1; Ptenfl/fl and Rosa26-LSL-SS18-SSX2; Ptenfl/fl mice [23] were maintained on a mixed strain background, C57BL/6 and SvJ. Mice were genotyped with the following primers: Rosa26-LSL-SS18-SSX (F flox—AAACCGCGAAGAGTTTGTCCTC, F wt—GTTATCAGTAAGGGAGCTGCAGTGG, R—GGCGGATCACAAGCAATAATAACC) Pten (F flox—CAAGCACTCTGCGAACTGAG, R—AAGTTTTTGAAGGCAAGATGC). TATCre was dosed with 10 μL intramuscular injections at 50 μM at 1 month of age. Then, animals were randomly divided into four groups; the control group receiving vehicle (PBS) (*n* = 7), jaspine B group receiving 5 mg/kg jaspine B (*n* = 4), empty liposome group receiving empty liposome without jaspine B (*n* = 3), and jaspine B liposome group receiving 5 mg/kg encapsulated jaspine B (*n* = 3) [23]. The tumor measurements were performed by an experienced person (blinded to the animal treatment group) using a digital caliper. Length, width, and height of the tumors were measured, and tumor sizes were calculated using a standard equation of W × D × L/2 and reported as mm^3^.

### 4.8. Statistical Analysis

Statistical analyses were performed using GraphPad Prism 8.0 statistical software (San Diego, CA, USA), and results were expressed as the mean ± SD. One-way analysis of the variances (ANOVA) was used to evaluate the differences between groups, followed by Tukey’s post hoc analysis. The level of significance was set at *p* < 0.05.

## 5. Conclusions

In this study, we formulated a liposomal drug delivery system for jaspine B and successfully adopted an LC-MS/MS method to characterize jaspine B in solution and liposomal formulation. Moreover, our in vitro and in vivo studies indicate that jaspine B is an effective sphingolipid metabolism inhibitor and a viable therapeutic option for treating sarcomas and other types of cancers. The remarkable 97% encapsulation of jaspine B in our novel liposome drug delivery system addresses low bioavailability issues, improves cancer cell uptake, and increases therapeutic efficacy to serve as a potential SS adjuvant therapy.

## Figures and Tables

**Figure 1 marinedrugs-20-00509-f001:**
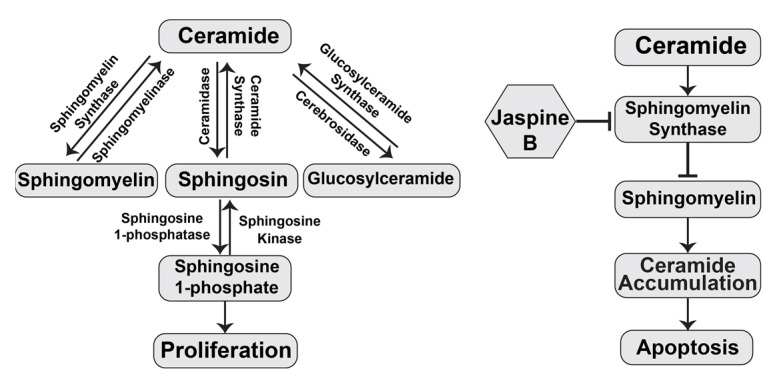
Schematic overview of the sphingomyelin pathway and jaspine B’s mechanism of action.

**Figure 2 marinedrugs-20-00509-f002:**
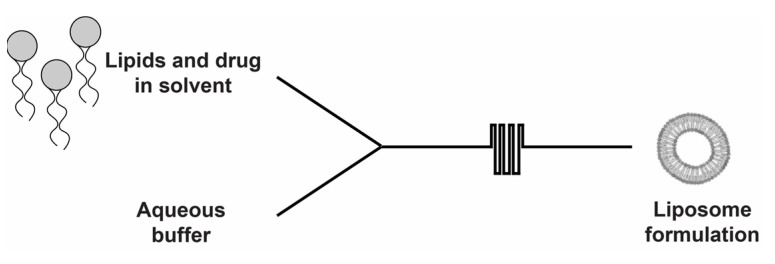
Schematic overview of the jaspine B liposome formation procedure using the microfluidics method.

**Figure 3 marinedrugs-20-00509-f003:**
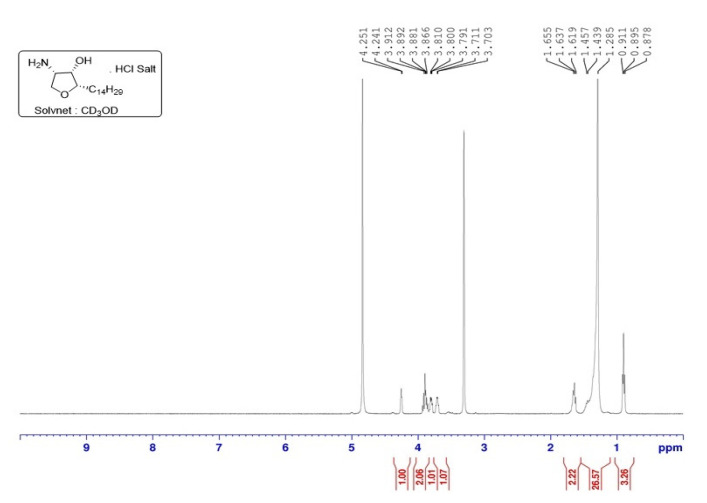
The ^1^H-NMR spectrum of scaled-up jaspine B HCl salt.

**Figure 4 marinedrugs-20-00509-f004:**
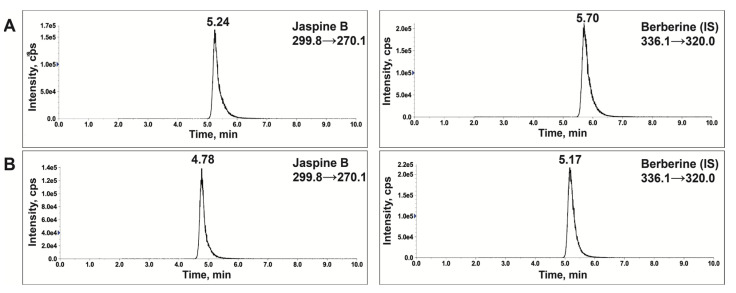
Respective multiple reaction monitoring (MRM) chromatograms of jaspine B and berberine as internal standard (IS) using LC-MS/MS. The chromatograms represent the MRM of jaspine B (*m*/*z* 299.8→270.1) and IS (*m*/*z* 336.1→320.0) from (**A**) standard solutions and (**B**) liposome samples.

**Figure 5 marinedrugs-20-00509-f005:**
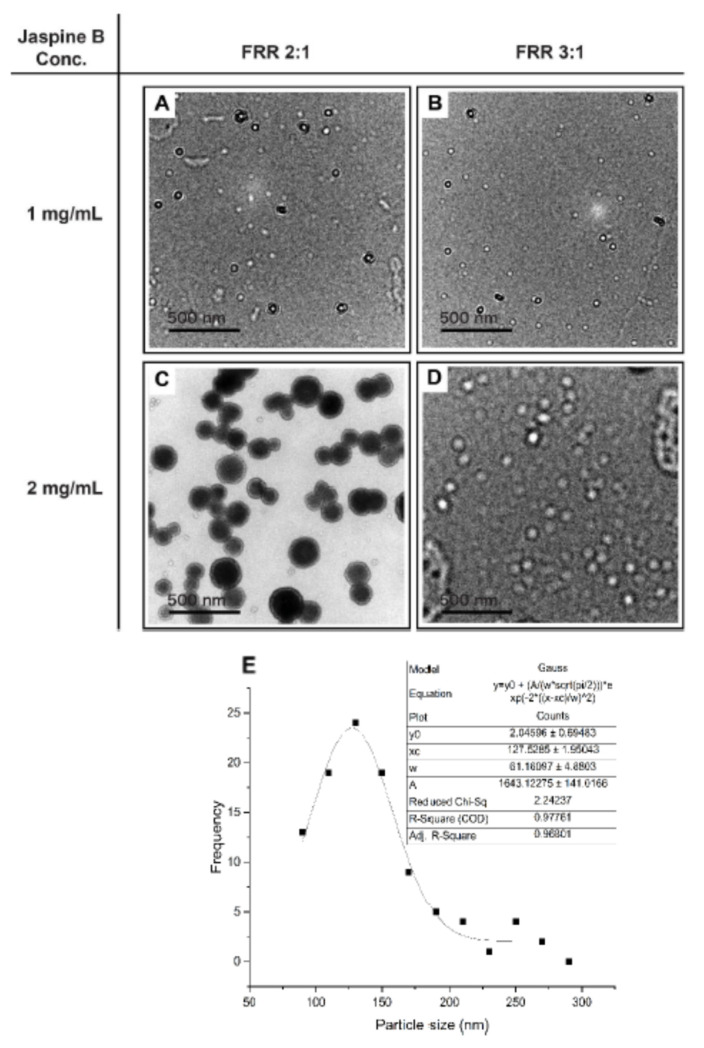
The TEM images of (**A**–**D**) jaspine B liposomes were formulated using microfluidics with different jaspine B concentrations and FRR. Liposomes with jaspine B concentration of 2 mg/mL and FRR 2:1 had the highest %EE. The size distribution of jaspine B liposome (**E**) with the highest %EE correspondence to TEM image C. The bar represents 500 nm. FRR; flow rate ratio, TEM; transmission electron microscopy, EE; encapsulation efficiency.

**Figure 6 marinedrugs-20-00509-f006:**
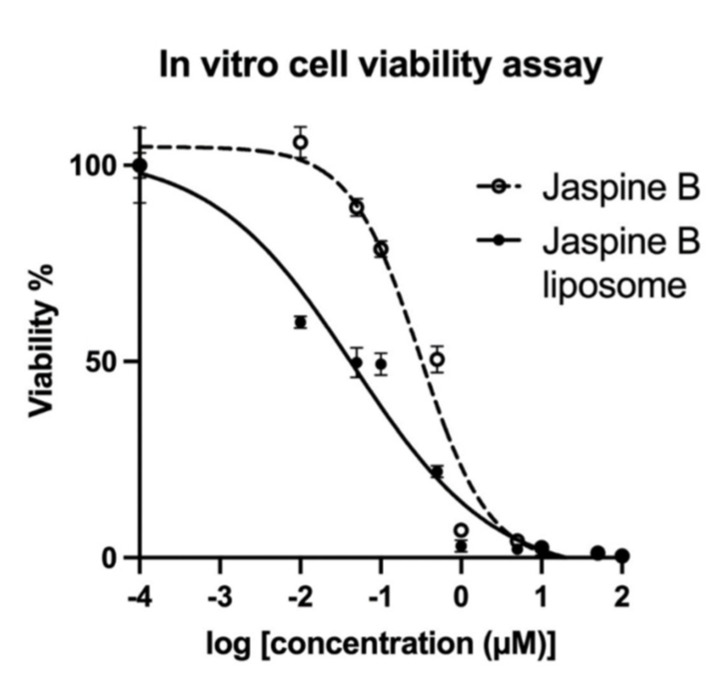
The cell viability percentage (normalized by drug-free medium or empty liposome-treated cells) of the Yamato-SS, a human synovial sarcoma cell line, was measured by MTT assay after treatment with final concentrations ranging from 0.01 to 100 µM of jaspine B or equivalent concentrations of jaspine B liposome and incubation for 72 h.

**Figure 7 marinedrugs-20-00509-f007:**
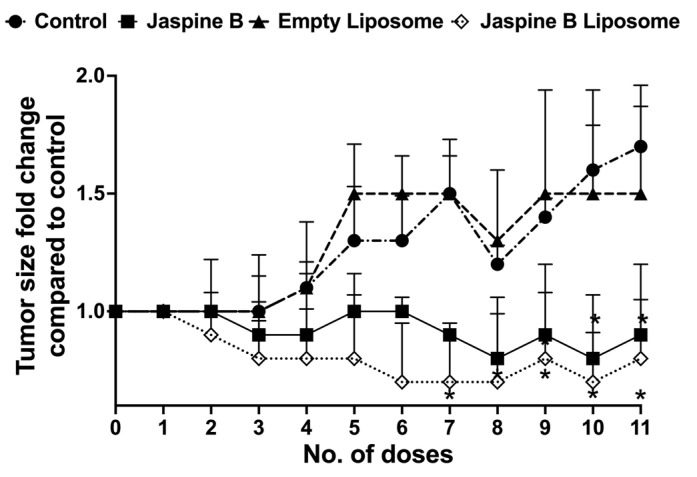
Tumor growth suppression in an animal model of synovial sarcoma (SS) after oral administration of three doses/week of the vehicle (PBS) in control (*n* = 7) or 5 mg/kg as jaspine B: jaspine B (*n* = 4), empty liposome (*n* = 3), or an equivalent dose of jaspine B in liposomal formulation (*n* = 3). * Significantly different from the control or empty liposome groups, *p* < 0.05.

**Figure 8 marinedrugs-20-00509-f008:**
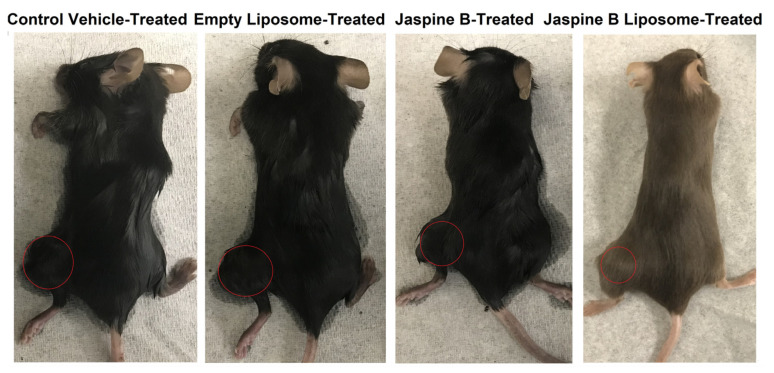
Representative images depict the tumor size in mice bearing SS tumor after treatment with 11 doses of vehicle, empathy liposome, jaspine B (5 mg/kg), or jaspine B liposome (an equivalent dose of jaspine B). The red circle highlights the location and the size of the tumor.

**Figure 9 marinedrugs-20-00509-f009:**
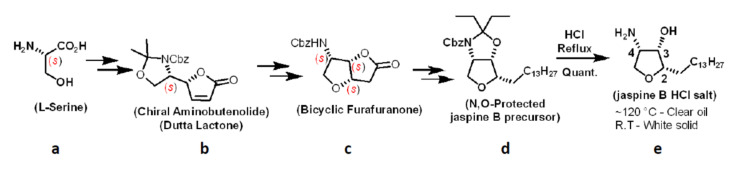
Stereoselective synthesis of jaspine B using chiral pool strategy. (**a**) L-Serine, (**b**) chiral aminobutenolide, (**c**) bicyclic furafuranone, (**d**) N, O-protected jaspine B precursor, (**e**) jaspine B as HCl salt.

**Figure 10 marinedrugs-20-00509-f010:**
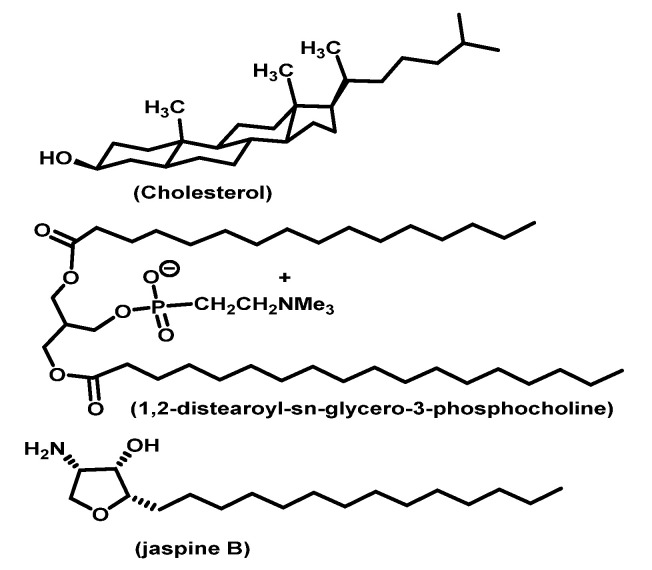
Amphiphilic chemical components of the liposomal formulation.

**Table 1 marinedrugs-20-00509-t001:** The effect of flow rate ratio (FRR) and jaspine B concentration on the liposome size distribution and encapsulation efficiency percentage (EE%).

Jaspine B Conc. (mg/mL)	FRR (PBS: Lipid)	Size (nm) Mean ± SD	EE%
1	2:1 3:1	100.1 ± 2.2 74.2 ± 7.2	70 43
2	2:1 3:1	127.5 ± 61.2 89.8 ± 3.7	97 77

**Table 2 marinedrugs-20-00509-t002:** IC_50_ and log IC_50_ values of jaspine B and its liposomes in the Yamato-SS cell line.

Drug	IC_50_ (µM) Mean ± SD	Log IC_50_ (µM) Mean ± SD
Jaspine B	0.36 ± 0.07	−0.44 ± 0.07
Jaspine B liposome	0.06 ± 0.01 *	−1.26 ± 0.07 *

The values are presented as mean ± SD. * Significantly different from jaspine B, *p* < 0.05.

## Data Availability

The data presented in this study are available on request from the corresponding author.

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
