# Peer review of "Formulation, Characterization, and In Vitro/In Vivo Efficacy Studies of a Novel Liposomal Drug Delivery System of Amphiphilic Jaspine B for Treatment of Synovial Sarcoma"

_marinedrugs, 2022, doi:10.3390/md20080509_

Round 1
Reviewer 1 Report
This manuscript reports a liposome nanoparticle containing a marine-derived drug Jaspine B for SS treatment. It presents the basic characterization data, such as size and cell toxicity and in vivo SS inhibition. Prior to acceptance, this manuscript needs a major revision. Some comments are provided below:
1. The mouse tumor treatment was stated to go on for 4 weeks. However, Figure 6 just shows 11 day data. Please make it consistent.
2. Figure 3: what is the size distribution and average size? What is the loading amount and loading efficacy of Jaspine B in the system? Is the loaded JB on the lipid bilayer or inside the liposome?
3. How stable is Liposome-JB? How quickly is JB released from liposome?
4. What is berberine? The derivative of Jaspine B?
5. Figure 5: why the cell viability was around 50% at 0.01 uM (the beginning point) for liposome-JB? What were EC50 values for two cases?
6. Figure 6: a. n=? b. is the dose of JB (5 mg/kg) the total injected dose or one injection dose? Please specify it.
7. In discussion, you should discuss why liposome-JB is more efficient than free JB, and then compare with other reports.
Author Response
We appreciated the reviewer's valid and constructive comments on our manuscript for further improving its quality. Please find the itemized response to each comment of the reviewer as follows.
This manuscript reports a liposome nanoparticle containing a marine-derived drug Jaspine B for SS treatment. It presents the basic characterization data, such as size and cell toxicity and in vivo SS inhibition. Prior to acceptance, this manuscript needs a major revision. Some comments are provided below:
- The mouse tumor treatment was stated to go on for 4 weeks. However, Figure 6 just shows 11-day data. Please make it consistent.
Answer: Thank you for noticing the mislabeling. The figure X-axes title is now fixed and is consistent with the text.
- Figure 3: what is the size distribution and average size? What is the loading amount and loading efficacy of Jaspine B in the system? Is the loaded JB on the lipid bilayer or inside the liposome?
Answer: The requested data are included in the text, Table 1, and Fig. 5B.
- How stable is Liposome-JB? How quickly is JB released from liposome?
Answer: Based on a short-period stability study, JB liposome is stable in PBS pH 7.4 at 4°C for more than two months and does not release quickly in PBS medium as the EE% was not significantly changed.
- What is berberine? The derivative of Jaspine B?
Answer: Berberine is a hydrophobic compound used as an internal standard in HPLC analysis. (https://doi.org/10.3390/md15090279).
- Figure 5: why the cell viability was around 50% at 0.01 uM (the beginning point) for liposome-JB? What were EC50 values for two cases?
Answer: The lower concentrations (empty liposome) was not included. The issue is fixed now, and the cell viability for both drugs starts at around 100%. The IC50 values are now added to the text and in Table 2 in the result section.
- Figure 6: a. n=? b. is the dose of JB (5 mg/kg) the total injected dose or one injection dose? Please specify it.
Answer: The number of animals per group is added to the text and figure 7. The 5 mg/kg dose was administered orally three times a week for four weeks (each administration contained 5 mg/kg of jaspine B or its equivalent dose as liposome).
- In discussion, you should discuss why liposome-JB is more efficient than free JB, and then compare with other reports.
Answer: The study findings are now discussed and compared with available literature.
Reviewer 2 Report
The manuscript entitled “Formulation, Characterization, and In vitro/In vivo Efficacy Studies of a Novel Liposomal Drug Delivery System of Amphiphilic Jaspine B for Treatment of Synovial Sarcoma” by Sana Khajeh pour et al have investigated encapsulated liposomes containing jaspine B.
The results are very preliminary. Further characterization of the delivery system needs to be performed. Additionally, scope of the in vitro characterization should be expanded to include penetration and cellular localization studies.
Some specific comments for revision:
1. In Methods is written “24 hours” and in the section 2.3 – “After 72 hours of exposure to serial dilutions…..”.
2. How authors can explain so high toxicity of jaspine B when in the other anti-tumoral studies of jaspine B the IC50 after 24 hours in the range from 2 to 10 µM (Ref: doi: 10.1194/jlr.M072611)?
3. Why curve of cytotoxicity of jaspine B liposome starts from 60% of viability?
4. As can be observed from Figure 5, the statistical analysis of the data was carried out for only one experimental point on the graph (50% of toxicity). The authors have to explain this fact.
5. For a better interpretation of the obtained data, in addition to the cytotoxicity Graph (Fig.5), the authors should present a table with the data of EC50 ±SD.
6. Why in the text of the section 2.4 is written “….four weeks of treatment.” and on the figure presented only 11 days?!
7. The introduction is rather generic and too short, I did not find references for the last studies of jaspine B. In the discussion a comparison to results of other studies on composition or cell viability or in vivo experiments of similar samples are lacking.
8. Reference 19 is not reported in the text
Considering the scope of this journal, the work is not substantial enough to be published in its current form.
Author Response
We appreciated the reviewer's valid and constructive comments on our manuscript for further improving its quality. Please find the itemized response to each comment of reviewers as follows.
The manuscript entitled "Formulation, Characterization, and In vitro/In vivo Efficacy Studies of a Novel Liposomal Drug Delivery System of Amphiphilic Jaspine B for Treatment of Synovial Sarcoma" by Sana Khajeh pour et al have investigated encapsulated liposomes containing jaspine B.
The results are very preliminary. Further characterization of the delivery system needs to be performed. Additionally, scope of the in vitro characterization should be expanded to include penetration and cellular localization studies.
Some specific comments for revision:
- In Methods is written "24 hours" and in the section 2.3 – "After 72 hours of exposure to serial dilutions…..".
Answer: The issue is fixed and the total exposure time was 72 hours.
- How authors can explain so high toxicity of jaspine B when in the other anti-tumoral studies of jaspine B the IC50 after 24 hours in the range from 2 to 10 µM (Ref: doi: 1194/jlr.M072611)?
Answer: We agree with the reviewer's point; however, the variation in the IC 50 value could be explained by the cell type and their sensitivity to the JB. JB's IC50 also has been reported to be at 186 nM, which is very close to our observation at a similar cell line (DOI: 10.1002/cbic.201800496)
- Why curve of cytotoxicity of jaspine B liposome starts from 60% of viability?
Answer: The lower concentrations (empty liposome) was not included. The issue is fixed now, and the cell viability for both drugs starts at around 100%. The IC50 values are now added to the text and in Table 2 in the result section.
- As can be observed from Figure 5, the statistical analysis of the data was carried out for only one experimental point on the graph (50% of toxicity). The authors have to explain this fact.
Answer: Thank you for the point. The statistical analysis was done for all the data points now.
- For a better interpretation of the obtained data, in addition to the cytotoxicity Graph (Fig.5), the authors should present a table with the data of EC50 ±SD.
Answer: The IC50 and log IC50 values are now presented in Table 2 and added to the text.
- Why in the text of the section 2.4 is written "…. four weeks of treatment." and on the figure presented only 11 days?!
Answer: Thank you for noticing the mislabeling. The figure X-axes title is now fixed and is consistent with the text.
- The introduction is rather generic and too short; I did not find references for the last studies of jaspine B. In the discussion a comparison to results of other studies on composition or cell viability or in vivo experiments of similar samples are lacking.
Answer: Thank you for the comment. The issue is fixed, and the study findings are now discussed and compared with available literature.
- Reference 19 is not reported in the text.
Answer: The issue is now fixed, and the reference list is updated.
Considering the scope of this journal, the work is not substantial enough to be published in its current form.
Answer: The manuscript is improved significantly by addressing the reviewers' comments.
Reviewer 3 Report
The presented manuscript is devoted to the development and study of the liposomal form of the Jaspine B drug and is certainly interesting to a wide range of readers of the journal. A very curious technique for preparing a liposomal form of a lipophilic drug using microfluidic technologies was used.
The manuscript is distinguished by conciseness and clarity of wording, however, before accepting it for publication, it is necessary to make several edits.
Nowhere in the text of the work is not substantiated the choice of lipid composition. The selection of a lipid composition for a new formulation is a non-trivial task, which can be devoted to a separate manuscript. These points should be clearly stated in the manuscript.
1. In the Introduction section, indicate the current status of the active molecule. Is it FDA approved? What delivery systems are currently in use?
2. Section Results, subsection 2.1. The authors indicate the optimal conditions for the preparation of liposomes, but are silent about how they came to this conclusion. I would like to see data (perhaps in SI) about what results were obtained under other experimental conditions. Also, according to microscopy data, the variety in the size of liposomes is large: from 100 to 300 nm. Here it would be very useful to obtain data from the DLS method and compare polydispersity indices. In general, the availability of data from microscopy certainly increases the level of research, and it is all the more unusual to observe the absence of such routine methods as DLS.
3. Subsection 2.2. It is necessary to clearly indicate what was the effectiveness of the inclusion of the drug, and what mass ratio of drug:lipid was achieved.
4. Subsection 2.3. Acute toxicity values ​​obtained for the control sample and liposomal formulation should be clearly stated. Details must be provided in the caption to Figure 5: cell line, experimental conditions, type of test (MTT or other).
5. Subsection 2.4. The choice of oral administration of the liposomal form of the drug is not obvious. Usually intravenous administration is used. Authors should justify their experimental design. The caption to figure 6 should indicate the number of groups of animals.
6. The Discussion section should be significantly expanded. Interesting data have been obtained from various methods and should be carefully considered and compared with the literature.
7. In section 4, subsections have the wrong first index. The subsection on synthesis has a typo in its title. If a synthesis was carried out in the work, it should be discussed in the Discussion section.
Author Response
We appreciated the reviewer's valid and constructive comments on our manuscript for further improving its quality. Please find the itemized response to each comment of reviewers as follows.
The presented manuscript is devoted to the development and study of the liposomal form of the Jaspine B drug and is certainly interesting to a wide range of readers of the journal. A very curious technique for preparing a liposomal form of a lipophilic drug using microfluidic technologies was used.
The manuscript is distinguished by conciseness and clarity of wording, however, before accepting it for publication, it is necessary to make several edits.
We appreciate the reviewer's positive notes on the article.
Nowhere in the text of the work is not substantiated the choice of lipid composition. The selection of a lipid composition for a new formulation is a non-trivial task, which can be devoted to a separate manuscript. These points should be clearly stated in the manuscript.
Answer: We agree with the reviewer’s comment and addressed it by including the following statement in the text. ”The lipid composition, their ratio, and jaspine B concentration were determined based on pilot study results, physicochemical characteristics of jaspine B, the manufacturer's protocol, initial suggestions from Precision NanoSystems scientists, and previous studies on lipophilic molecules [20-22].”
- In the Introduction section, indicate the current status of the active molecule. Is it FDA approved? What delivery systems are currently in use?
Answer: The statement “Jaspine B is still in the investigational phase and currently there is no commercial product available for therapeutic use as a natural product.”
- Section Results, subsection 2.1. The authors indicate the optimal conditions for the preparation of liposomes, but are silent about how they came to this conclusion. I would like to see data (perhaps in SI) about what results were obtained under other experimental conditions. Also, according to microscopy data, the variety in the size of liposomes is large: from 100 to 300 nm. Here it would be very useful to obtain data from the DLS method and compare polydispersity indices. In general, the availability of data from microscopy certainly increases the level of research, and it is all the more unusual to observe the absence of such routine methods as DLS.
Answer: The issue is addressed and the characterization data is included in the text as a table and figure. However, due to a lack of access to DLS, we analyzed several TEM images to acquire the size distribution of the JB liposomes.
- Subsection 2.2. It is necessary to clearly indicate what was the effectiveness of the inclusion of the drug, and what mass ratio of drug:lipid was achieved.
Answer: The issue was addressed in the text. Considering an EE of 97% the drug:lipid mass ratio was 0.39.
- Subsection 2.3. Acute toxicity values ​​obtained for the control sample and liposomal formulation should be clearly stated. Details must be provided in the caption to Figure 5: cell line, experimental conditions, type of test (MTT or other).
Answer: The acute toxicity values are now presented in Table 2 and added to the text and the suggested information is added to the figure 6 caption.
- Subsection 2.4. The choice of oral administration of the liposomal form of the drug is not obvious. Usually intravenous administration is used. Authors should justify their experimental design. The caption to figure 6 should indicate the number of groups of animals.
Answer: Thank you for the valid comment. The oral administration of the liposomal is justified in the introduction and discussion sections.
- The Discussion section should be significantly expanded. Interesting data have been obtained from various methods and should be carefully considered and compared with the literature.
Answer: The study findings are now discussed and compared with available literature.
- In section 4, subsections have the wrong first index. The subsection on synthesis has a typo in its title. If a synthesis was carried out in the work, it should be discussed in the Discussion section.
Answer: The issue is fixed now. The JB synthesis has been reported before and we just scaled it up and some conformational data added to the manuscript.
Round 2
Reviewer 1 Report
The authors have carefully addressed the questions and concerns raised by the reviewers and made extensive revision and improvement.
Probably some editings and language expressions may be needed to improve.
Author Response
We appreciated the reviewer's positive comments on the manuscript revision and are pleased to know that all points have been addressed appropriately.
The manuscript was subjected to an English edition by a professional editor to address the reviewer's last comments.